# Inference of gene regulation functions from dynamic transcriptome data

Patrick Hillenbrand[1], Kerstin C Maier[2], Patrick Cramer[2]*, Ulrich Gerland[1]*

[1]Lehrstuhl für Theorie komplexer Biosysteme, Physik-Department, Technische Universität München, Garching, Germany; [2]Max-Planck Institute for Biophysical Chemistry, Göttingen, Germany

**Abstract** To quantify gene regulation, a function is required that relates transcription factor binding to DNA (input) to the rate of mRNA synthesis from a target gene (output). Such a 'gene regulation function' (GRF) generally cannot be measured because the experimental titration of inputs and simultaneous readout of outputs is difficult. Here we show that GRFs may instead be inferred from natural changes in cellular gene expression, as exemplified for the cell cycle in the yeast *S. cerevisiae*. We develop this inference approach based on a time series of mRNA synthesis rates from a synchronized population of cells observed over three cell cycles. We first estimate the functional form of how input transcription factors determine mRNA output and then derive GRFs for target genes in the *CLB2* gene cluster that are expressed during G2/M phase. Systematic analysis of additional GRFs suggests a network architecture that rationalizes transcriptional cell cycle oscillations. We find that a transcription factor network alone can produce oscillations in mRNA expression, but that additional input from cyclin oscillations is required to arrive at the native behaviour of the cell cycle oscillator.

*For correspondence: patrick.
cramer@mpibpc.mpg.de (PC);
gerland@tum.de (UG)

Competing interest: See
page 19

Reviewing editor: Sarah A
Teichmann, EMBL-European
Bioinformatics Institute &
Wellcome Trust Sanger Institute,
United Kingdom

## Introduction

Much of the topology of the yeast transcriptional network is known from functional genomics approaches such as chromatin immunoprecipitation and mRNA expression data (*Lee et al., 2002*; *Harbison et al., 2004*; *Tsai et al., 2005*; *Wu et al., 2006*; *Hu et al., 2007*). However, the nature of the nodes in such networks, where the input signals are integrated into a transcriptional response, remain elusive. The quantitative design of a network node is known as 'gene regulation function' (GRF) and is of central importance for understanding the regulatory dynamics of the network. In *E. coli*, the measurement of a GRF was demonstrated for a single node with two inputs (*Setty et al., 2003*; *Kuhlman et al., 2007*). However, the approach relies on specific inducers for the involved transcription factors and is applicable only to a very limited set of genes.

GRFs must therefore be inferred from indirect experimental evidence. In multicellular organisms, GRFs for developmental genes can be inferred from the spatially varying profiles of morphogens (*Jaeger et al., 2004*; *Segal et al., 2008*; *Junker et al., 2014*). For a single-cell organism such as yeast, the reconstruction of GRFs must instead rely on the temporal variation of input factors. The variation can either be intrinsic, as observed during the cell cycle (*Spellman et al., 1998*), or it can be triggered by an external perturbation, as during DNA damage (*Workman et al., 2006*) or osmotic stress (*Miller et al., 2011*). A major obstacle for inferring GRFs is that GRFs describe gene activity but expression data typically provide only total mRNA levels. This limitation is overcome by 'Dynamic Transcriptome Analysis' (DTA), which uses nonperturbing metabolic RNA labeling to additionally obtain the amounts of newly synthesized mRNA, which can be used as a proxy for RNA synthesis rates and gene activity (*Miller et al., 2011*; *Sun et al., 2012*).

**eLife digest** Living cells rely on networks of genes to control their behavior, including how they grow, develop and respond to stress. Genes encode instructions needed to make proteins and other molecules, and much of the control is exerted at the first stage of protein production, known as transcription. During this process, a gene is copied to make molecules known as transcripts that may later be used as templates to make proteins.

Many genes encode proteins that act to regulate transcription. Therefore, an individual gene may receive inputs from other genes, and these inputs affect how much transcript the gene produces, which can be considered as the gene's output. While these inputs and outputs can often be wired together to form a network, it is less clear exactly how all the different inputs at a gene interact to determine its output. These interactions are known as "gene regulation functions", and knowing them would be an important step towards understanding gene networks, which would help us to predict how cells will behave in different situations.

Gene regulation functions are difficult to measure directly, so researchers would like to find other ways to assess them indirectly. A recently developed experimental technique called "dynamic transcriptome analysis" seemed promising as it measures both the inputs and outputs of all genes in a cell over time. Hillenbrand et al. used this technique to infer gene regulation functions with one or two inputs in yeast cells.

Comparing these estimates with experimental data from previous studies showed that these inferred gene regulation functions could successfully predict the output of a gene based on its inputs. Hillenbrand et al. then used these estimates to search and model a well-known genetic network that is thought to be part of the molecular clockwork that controls the timing of events that cause a cell to divide.

Currently, the approach used by Hillenbrand et al. treats gene regulation functions like "black boxes". This means that, while an output can be predicted if the inputs are known, it cannot reveal all of the detailed mechanisms behind it. Gaining insights into the inner workings of these black boxes will require taking more data into account, such as how abundant the proteins that regulate transcription are, where they are located within cells or whether they are active or not. Therefore, the next challenge is to incorporate these kinds of data to gain a fuller picture of how gene networks operate within cells.

Cell cycle progression in yeast is controlled by cyclin-dependent kinases (CDKs) that are activated by periodically expressed cyclins (*Evans et al., 1983*). In addition, periodic transcription occurs in waves (*Breeden, 2000*; *Breeden, 2003*) and plays a role in maintaining the cell cycle (*Wittenberg and Reed, 2005*). Several periodically expressed transcription factors control each other, forming a regulatory module (*Simon et al., 2001*). The observation of periodic gene expression even in the absence of mitotic cyclins indicated that sequential transcription activation is sufficient for periodic expression of most cell cycle genes (*Orlando et al., 2008*; *Haase and Reed, 1999*). However, coupling to cyclin-CDK activity serves as a pacemaker and increases robustness of the transcriptional oscillator (*Simmons Kovacs et al., 2012*). Several models for a minimal transcriptional cell cycle oscillator have been proposed (*Simon et al., 2001*; *Lee et al., 2002*; *Orlando et al., 2008*; *Sevim et al., 2010*; *Simmons Kovacs et al., 2012*). Parts of the transcriptional network were also reconstructed computationally from expression data (*Chen et al., 2004*; *Wu et al., 2006*). However, a comprehensive understanding of the transcriptional cell cycle network is still missing, and the degree of its dependence on the cyclin oscillator is debated (*Bristow et al., 2014*; *Rahi et al., 2016*).

Here we develop a method to infer GRFs from DTA data and apply it to cell cycle genes in yeast. We use DTA data providing the mRNA synthesis rates and levels for synchronized *S. cerevisiae* cells over three cell cycles in two replicate experiments (*Eser et al., 2014*). We consider target genes with significant regulatory inputs from one or more transcription factors. We restrict our analysis to cases where evidence for physical interaction between transcription factors and target genes exists or a genetic interaction is established. We apply our method to infer GRFs of cell cycle regulated

transcription factors. We deduce possible models for a transcriptional cell cycle oscillator and test their capability to generate oscillations without cyclin-CDK activity. Our approach may be extended to quantitatively describe other gene regulatory systems, such as stress response mechanisms, apoptosis, or cell differentiation networks.

## Results

### Inference of gene regulation functions

Our method to infer gene regulation functions (GRFs) from DTA data is illustrated in *Figure 1*. After selecting a target gene of interest, we compile a list of known input factors and focus on those that display a significant fold-change in mRNA level over the time course of the experiment. We assume that their dynamics can rationalize the output dynamics (*Figure 1B*) via a smooth input-output relation, the GRF. This assumption is viable even for genes that belong to a larger regulatory network. We can treat each gene independently because the DTA data provide mRNA time traces $m(t)$ for all input factors and output mRNA synthesis rates $s(t)$ for most genes (*Figure 1A*). The inputs may be transcription factors or cofactors (*Siggers et al., 2011*), but for simplicity we refer to all inputs as transcription factors (TFs). Here, we do not explicitly consider post-transcriptional regulation of TFs or potential inputs from regulatory RNAs.

We infer a GRF by constructing a parameterized model, which describes the measured target gene output $s(t)$ via Hill-type functions of the input TF levels (*Figure 1*, 'quantitative model'). For the case of a single input, the GRF is parameterized by the basal activity $b$, the regulation amplitude $\alpha$, the response threshold $K$, and the sensitivity $n$. The parameters are estimated by finding the global best fit of our model to the measured target gene activity using the simulated annealing technique, see 'Materials and methods'. The effect of the input TFs can be either activating or repressing. To discriminate between these possibilities, we fit models for both and select the one that yields the better score.

Because protein levels are unknown, we infer a proxy $p(t)$ for the TF concentration from its mRNA level (see box 'quantitative model' in *Figure 1*) using a minimal model with a constant mRNA translation rate $\nu_t$ and a constant effective protein degradation rate $\lambda$. The resulting $p(t)$ is therefore both delayed and smoothened with respect to the input mRNA concentration $m(t)$. The characteristic timescale for both the delay and the smoothening of the signal is the effective protein half-life $\tau_{1/2} = \ln(2)/\lambda$. *Figure 1C* visualizes the effects of different protein degradation rates, and shows that information about the parameter $\lambda$ for a TF is contained in the transcription rate trajectories of target genes: only for an appropriate choice of $\lambda$ does the target gene activity $s(t)$ plotted against $p(t)$ collapse to a curve, which corresponds to the GRF of the target gene. This data collapse also serves as a visual consistency check for our approach. Note that besides the actual degradation of proteins, the timescale $\tau_{1/2}$ subsumes the effects of a number of molecular processes that are not yet characterized quantitatively. For instance, transport into and out of the nucleus, phosphorylation and dephosphorylation of transcription factors, and dilution of protein levels by cell growth all affect how the levels of activated transcription factors 'seen' by their target genes dynamically adapt after their mRNA level has changed. Effectively, these processes create a time delay, which is captured in our coarse-grained model by the single parameter.

The loss of synchrony within the measured cell population does not significantly impact the ability of our model to infer GRFs. This is exemplified by the TF Swi4 and its target gene *Rnr1* (*Figure 1*). Both *Swi4* and *Rnr1* are periodically expressed with a period of ~60 min, corresponding to the cell cycle period of the used yeast strain under the conditions of the experiments. The loss of synchrony of cell cycle progression between different cells is observed as a dampening of the oscillations, due to the population average over cells that increasingly diverge in their relative cell cycle phase. However, this dampening occurs for inputs and outputs alike, and our nonlinear inference scheme is tolerant against a partial loss of synchrony: The limited cell-to-cell variation in TF levels at a given time point samples only a limited regime of the GRF input range, within which the nonlinear GRF appears locally linear and is therefore not significantly affected by the population average. In the following we focus our analysis mainly on cell cycle-dependent genes, since most genes with significant fold-change in our dataset are periodically expressed.

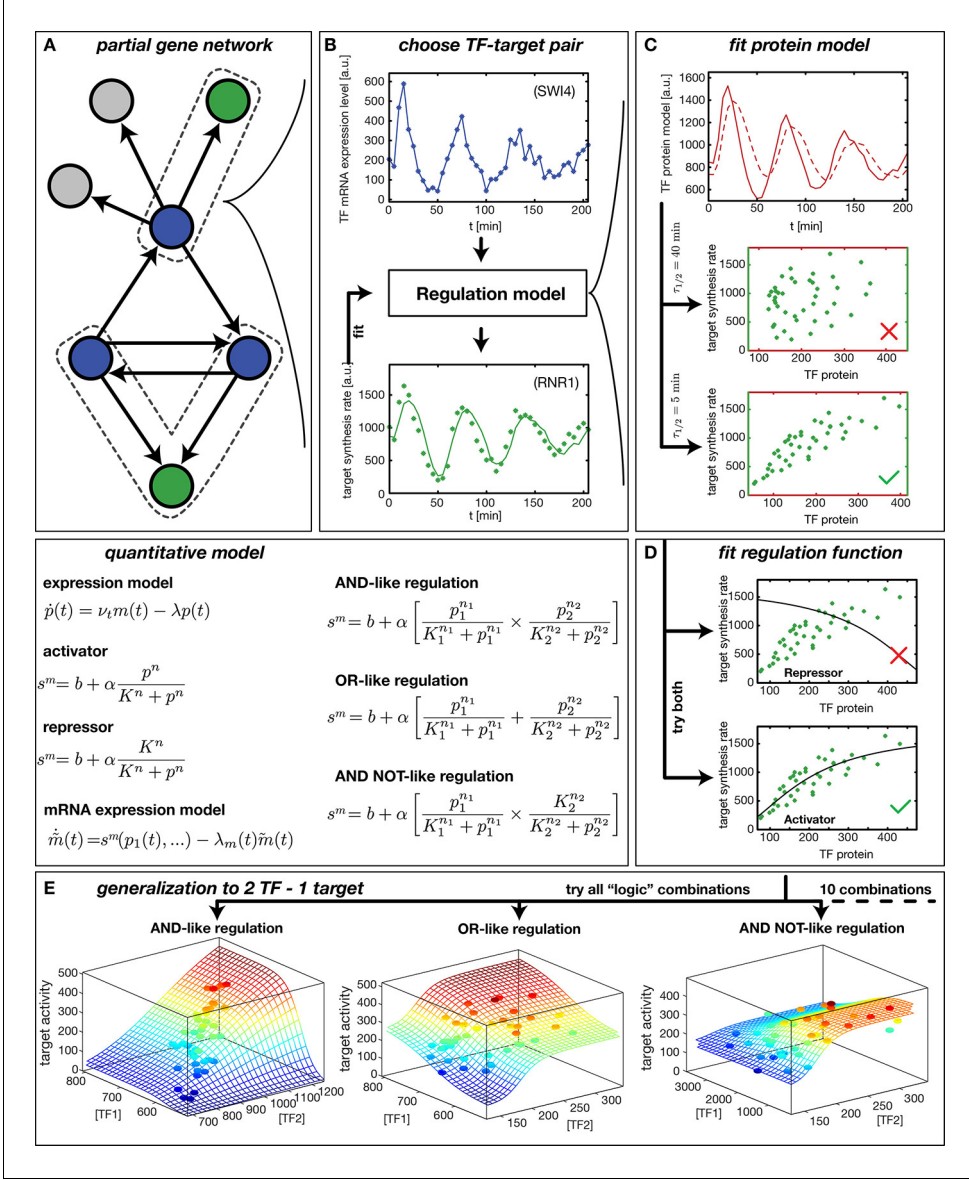

**Figure 1.** Reconstruction of regulation functions. (**A**) We select a target gene (green) and TFs (blue), which are known from the literature to interact with the target gene. We consider only target genes and corresponding TFs, which show significant variation in our dataset. (**B**) The fitting task is to find a regulation model using the TF mRNA expression level (upper plot), so that the time series of the target synthesis rate (dots in lower plot) is best described by the model synthesis rate $s^m$ (solid curve in lower plot). (**C**) We describe the concentration of active TF protein $p(t)$ by a simple model with translation rate $\nu_t$ and linear degradation rate $\lambda$ (see box quantitative model). For a correctly chosen $\lambda$ the target gene synthesis rate plotted against the TF protein concentration collapses to a curve (bottom plot). (**D**) Together with $p(t)$ a regulation function (target synthesis rate $s^m$ as a function of TF protein $p$) is estimated. For a single TF the regulation function has four parameters $(b, \alpha, K, n)$ and two possible directions of regulation: activator and repressor (see box mathematical model). To find the right regulatory direction we fit both and select the one, which yields the better score. (**E**) Two TFs can interact in multiple ways to generate a two dimensional combinatorial regulation function (**Buchler et al., 2003**). For two TFs we estimate the protein models and six parameters for the regulation function. There are 10 non-trivial regulatory analog 'logic' operations to test, of which 3 examples are depicted (see box quantitative model for the corresponding equations).

Our method is applicable not only to genes with single input signals, but also to some cases of combinatorial regulation. We take the combinatorial interaction of multiple input factors into account by inferring the best fitting gene regulatory 'logic' (*Buchler et al., 2003*) along with the parameters that characterize the shape of the GRF. We combine activating and repressing Hill functions to model analog combinatorial 'logic' response functions (*Figure 1*). For example, two activating TFs can up-regulate a target gene either individually (*Figure 1E*, 'OR logic') or cooperatively, if both are abundant (*Figure 1E*, 'AND logic'). For two input TFs we fit ten different non-trivial GRFs and select the 'logic' function with the best score; see 'Materials and methods' for the complete list of the types of GRFs that we consider. The score is defined as the fraction of the variance in the data that is not explained by the model, see 'Materials and methods'. Our fits minimize the score as a function of the GRF parameters.

## Illustration of GRF inference at individual target genes

To illustrate and test our method, we focused on the *CLB2* cluster, which comprises a set of genes that are expressed during the G2/M phase of the cell cycle (*Spellman et al., 1998*). We considered the input factors Fkh2, Ndd1, and Fkh1. Fkh2 binds to a consensus regulatory DNA element as a complex with Mcm1 (*Spellman et al., 1998*; *Zhu et al., 2000*). The Fkh2-Mcm1-DNA complex recruits the coactivator Ndd1 (*Koranda et al., 2000*; *Darieva et al., 2003*). Fkh1 plays a complementary and partially redundant role (*Kumar et al., 2000*; *Hollenhorst et al., 2000*; *Hollenhorst et al., 2001*). Because Mcm1 is not significantly periodically transcribed (*Figure 2—figure supplement 1*), we treat only Fkh2, Ndd1, and Fkh1 as significant regulatory inputs.

*Figure 2A and B* show the results of our analysis for two target genes in the *CLB2* cluster, *Hof1* and *Mob1*, which are not regulated by Fkh1 (*Harbison et al., 2004*; *Tuch et al., 2008*; *Zhu et al.,*

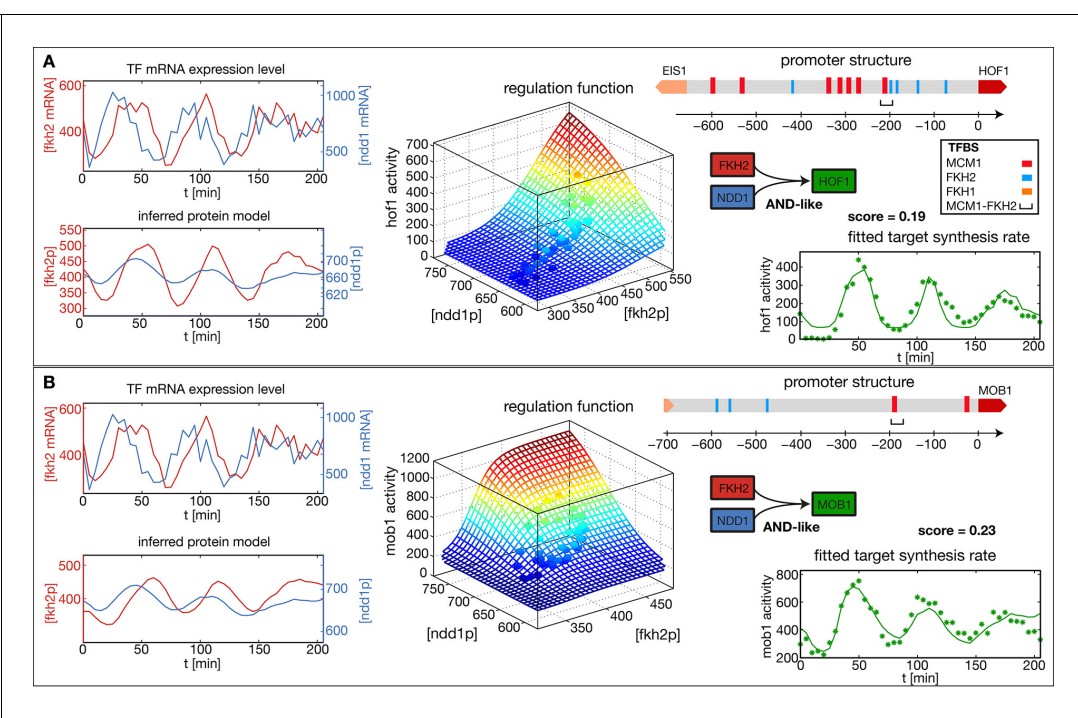

**Figure 2.** Reconstructed regulation functions of two genes in the *CLB2* cluster. For both target genes the measured TF expression level, the inferred protein level, the estimated regulation function with corresponding measured data points and the fit to the target synthesis rate are shown. Additionally predicted binding sites for relevant TFs within the target gene promoter regions are depicted. Both target genes, *Hof1* and *Mob1*, are regulated by Fkh2 and Ndd1 (which is recruited to the complex Mcm1-Fkh2).

The following figure supplement is available for figure 2:

**Figure supplement 1.** Non-periodic cell cycle regulators.

*2000*). In each case, the promoter structure is shown in the top right. TF binding sites and consensus motifs of the *CLB2* cluster regulatory element are indicated as obtained by motif search in the upstream sequences of each target gene (see 'Materials and methods' for details). The mRNA level time series for the input factors Fkh2 and Ndd1 are shown in the top left of each panel, with the inferred proxies $p(t)$ for the corresponding protein levels shown below. Here, the $p(t)$ time series for the same input TFs have been independently determined from the fits to the two target gene output time series. As a consequence, the resulting $p(t)$ time series for the same TF can deviate from each other. This illustrates the uncertainty in the inference of the parameter $\lambda$ from the output of an individual target gene. However, if several target genes of the same TF are known, then a mutually consistent estimate of $\lambda$ can be determined. Since all candidate TFs for the proposed transcriptional cell cycle oscillator that we study below have multiple target genes, we describe a method for the unified inference of $\lambda$ in the next section.

The estimated GRFs in *Figure 2* are depicted as surface plots, where the mesh represents the best-fit mathematical function within our set of GRF types and the discrete points show which parts of the GRF are sampled by the experimental data. The corresponding time series of the output promoter activity is shown in the bottom right of each panel, including both the model output (continuous line) and the experimental data (points). As can be seen from these plots, the inferred GRFs capture the expression dynamics of the target genes relatively well (best fit scores, i.e. the remaining fraction of the variance in the data not explained by the model, are indicated in the figure). In both cases, the estimation procedure yields an AND-like logic for the action of the input TFs Fkh2 and Ndd1, such that full activation can be obtained only in the presence of both inputs. For the target gene *Hof1*, however, the data points sample only a narrow region in the two-dimensional Ndd1-Fkh1 concentration space of the GRF, such that the shown two-dimensional GRF is an extrapolation that cannot be verified with this data set. This example illustrates a useful property of our GRF inference scheme: the sampling density in the input space of the GRF already indicates the range over which the GRF inference is supported by the data.

The *CLB2* cluster target gene *Kip2* provides an example for a complex GRF inference task, since it has Fkh1 as regulatory input in addition to Fkh2 and Ndd1 (*Harbison et al., 2004*). Given the combinatorial complexity of 3-input GRFs and the limited amount of expression data, we cannot simply extend our approach to simultaneously treat more than two inputs. Instead, it is necessary to limit our inference to the two most significant inputs. We used *Kip2* as a test case for a method to identify the most significant inputs. As a reference, we first constructed a thermodynamic regulation model (*Bintu et al., 2005*)for *Kip2* based on its promoter structure (*Figure 3A*). This physico-chemical model is more detailed and requires more parameters than the Hill-type functions. It parameterizes the GRF by the TF affinities to their DNA binding sites, the interactions between TFs bound to proximal or overlapping sites, and the attraction of bound TF complexes towards RNA polymerase, see *Figure 3A* (top right). We estimated the parameter values from the data with the same method as used above (see 'Materials and methods'). The resulting 3-input GRF yielded a good description of the output (*Figure 3A*, bottom right) and predicted that the input from Fkh2 has only a minor effect.

To test whether our general inference method would yield the same conclusion without constructing a physico-chemical promoter model, we applied our method for 2-input GRFs to each combination of two input factors out of Fkh1, Fkh2, and Ndd1. Out of these combinations an AND-like GRF with Fkh1 and Ndd1 as the input obtained the best score (*Figure 3B*), consistent with the prediction that Fkh2 has the least significant effect. Notably, the obtained fit to the time-dependent output is almost identical to the one obtained with the thermodynamic model (bottom right panel of *Figure 3A and B*, respectively), despite the smaller number of fitted parameters in the general inference method (8 instead of 12 parameters). When the GRF of the thermodynamic model is plotted as a function of Fkh1 and Ndd1 with Fkh2 fixed at its average value (*Figure 3A*, center) the resulting projection is also very similar to the GRF of *Figure 3B*. Our analysis is therefore consistent with the interpretation of Ndd1 as the limiting factor in the formation of the Mcm1-Fkh2-Ndd1 DNA-bound complex in the *Kip2* promoter during the cell cycle.

The examples of *Figures 2* and *3* illustrate that our method to infer GRFs not only recapitulates known roles of TFs on their target genes, but also provides additional quantitative insight into the individual or combinatorial effects of TFs. Our analysis of *Kip2* regulation suggests that the two most significant regulatory inputs to a gene can be identified by applying the inference method for 2-input GRFs to each combination of two input factors and choosing the combination that yields the

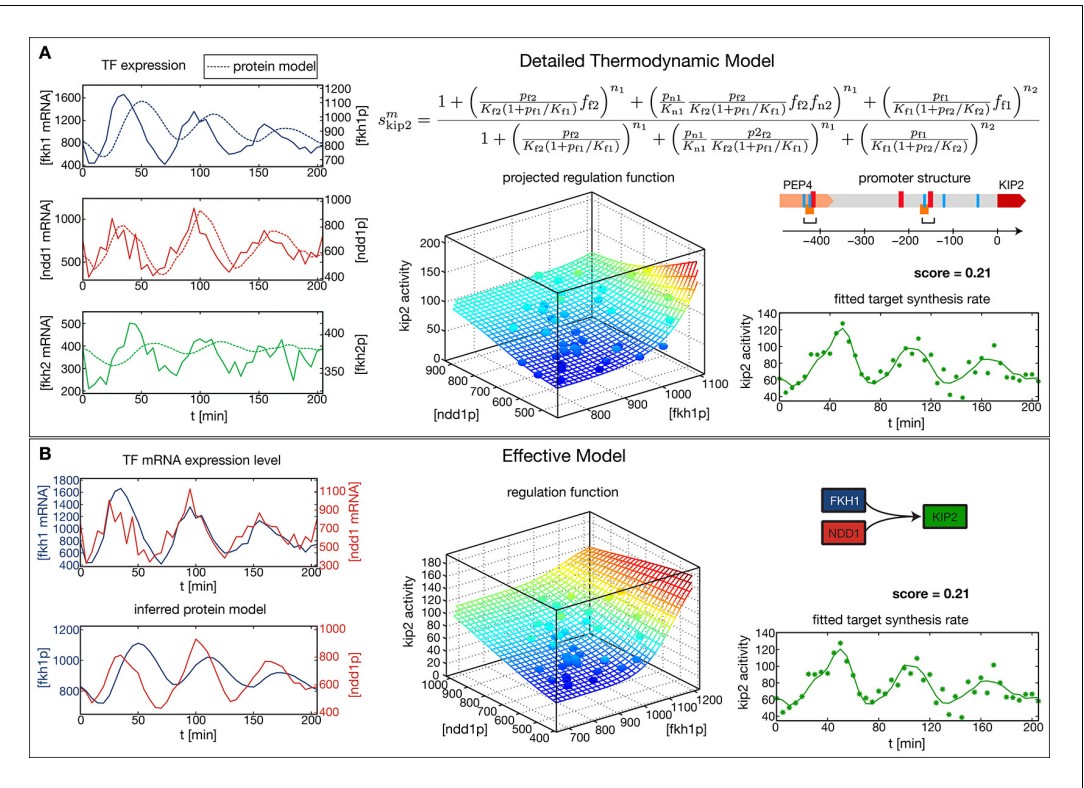

**Figure 3.** Comparison of the effective regulation function to a detailed promoter model. Target gene *Kip2* within the *CLB2* cluster is predominantly regulated by three input factors Fkh1, Fkh2 and Ndd1. (**A**) Results of fitting a detailed promoter model, using all three input factors. The regulation model $s_{\text{kip2}}^m$ is derived from the promoter structure which contains overlapping binding sites for Mcm1-Fkh2 and Fkh1, using a thermodynamic approach as detailed in *Bintu et al. (2005)*. The displayed projected regulation function has been plotted with Fkh2 fixed at its average value which leads to only minor deviations from the full model. (**B**) Results for fitting an effective model to *Kip2*. All three combinations of two out of three input factors have been tested with Fkh1/Ndd1 yielding the best score.

The following figure supplements are available for figure 3:

**Figure supplement 1.** Validation of our inferred GRFs with an independent dataset.

**Figure supplement 2.** Comparison of protein space coverage.

best score. In general, the relative significance of the different regulatory inputs to a gene will depend on the physiological conditions, and the method can only infer a relative significance for the conditions of the experiment. So far our analysis was limited to our calibration dataset (*Eser et al., 2014*) from which we infer our GRFs. To probe the consistency of our GRFs with independent data, and to test their ability to predict regulatory effects outside the regime of their calibration, we considered another published dataset that was measured under different, albeit similar, experimental conditions (*Pramila et al., 2006*). We asked whether the same GRFs that we inferred from the data of *Eser et al. (2014)* would also be able to describe the mRNA dynamics of this dataset, which we had not used to calibrate our GRFs.

*Pramila et al. (2006)* provide microarray data from $\alpha$-Factor synchronized yeast cells, which follows transcript levels over two cell cycles at 5 min intervals. We used rescaled TF expression data from this test dataset (see 'Materials and methods') as input for the GRFs inferred from our data. We then compared the output of our GRFs with the mRNA time series of the target genes in the test data. For this comparison, we used the GRFs and effective protein half-lives that we inferred for our analysis of transcriptional cell cycle oscillators presented in the next section. Hence, the output

curves are predictions without fit parameters, and the good agreement achieved in the comparison with the test data (*Figure 3—figure supplement 1*) suggests that our GRFs have predictive power beyond the regime of their calibration. *Figure 3—figure supplement 2* shows that the regime of input TF concentrations that is effectively sampled by the test data is similar but not identical to the range in our calibration data set.

It would also be desirable to be able to make in silico predictions of mutant behaviors with the inferred GRFs. For instance, a TF knockout could be emulated by setting the input from that TF to be zero for the GRFs of its target genes. However, other inputs of the target genes may also be affected by the knockout, leading to indirect effects on the expression pattern. Therefore, prediction of mutant behaviors will generally require a complete model of the genetic module in which that target gene is embedded. We discuss the inference of genetic modules in the next section, and return to the question of mutant behaviors further below.

## Inference of genetic modules

We next applied our method to multi-gene systems. As an illustration we wanted to test whether the data of *Eser et al. (2014)* is consistent with a proposed transcriptional cell cycle oscillator (*Haase and Reed, 1999*; *Orlando et al., 2008*; *Simmons Kovacs et al., 2012*). While the cell-cycle dependent oscillation in the expression of key cell-cycle TFs is clearly established, the underlying regulatory network is not comprehensively characterized. Furthermore, it is debated whether these TFs form a genuine transcriptional oscillator that can create transcriptional oscillations even without a functional protein-level cyclin oscillator (*Bristow et al., 2014*; *Rahi et al., 2016*). By applying our inference method to the dynamic transcriptome data of synchronized wild-type cells, we can test whether the oscillatory transcription profiles of a selected set of TFs form a consistent dynamic system of mutually regulating genes. To that end, we first pursued an autonomous transcriptional oscillator but included coupling to the cyclin-CDK oscillator where it turned out to be necessary for consistency with our data.

Rather than basing our analysis on specific network architectures (*Simon et al., 2001*; *Simmons Kovacs et al., 2012*), we decided to systematically generate and rank all plausible networks according to their global consistency with the DTA data. The challenge of such an unbiased approach is the combinatorial explosion of the number of core networks that can be generated even from small numbers of TFs when each of them has several possible regulatory interactions. Due to this combinatorial explosion, any attempt to first construct all possible networks and then fit each of them to the dynamic transcriptome profiles of all involved genes would necessarily fail. We therefore took a stepwise approach, in which we pre-fitted all possible combinations of at most two inputs to each node of the network, allowing us to then construct and evaluate all possible networks in a combinatorial scheme. As illustrated in *Figure 4*, our complete approach involves four steps: (A) selection of candidate TFs, (B) construction of their interaction matrix, (C) inference of a unified protein proxy for each TF in the set using different target genes, and (D) inference of all possible GRFs and combination into candidate networks, which are ranked according to their compatibility with the DTA data. The final step overcomes the combinatorial complexity by exploiting the modularity of the problem. In the following, we briefly provide essential information for each of the steps; additional details are provided in 'Materials and methods'.

For the selection of candidate TFs, we imposed the following criteria (*Figure 4A*): (i) distinct periodic expression in the DTA data, (ii) control by at least one other TF within the set, and (iii) at least one regulatory target within the set. The criteria (ii) and (iii) are necessary to obtain closed (strongly connected) networks. We focused on closed networks in order to identify possible core motifs for an autonomous transcriptional oscillator. However, we will see below that a consistent interpretation of the data requires additional input from the cyclin-CDK oscillator. To quantify the degree of periodicity of gene expression profiles, we relied on a previously established method (*Eser et al., 2014*). We considered the top 500 periodic genes, out of which 20 were TFs. We then used the direct interactions between these genes documented in the YEASTRACT database (*Teixeira et al., 2006*) to apply our criteria (ii) and (iii). This resulted in the following set of 11 TFs (see 'Materials and methods' for details): Hcm1, Swi5, Ace2, Yhp1, Swi4, Fkh1, Fkh2, Ash1, Yox1, Fhl1, and Ndd1. The factors Swi5 and Ace2 are known to be homologous (*Doolin et al., 2001*) and display synchronous expression in the DTA data set. We therefore considered them as a single network node for the purpose of our

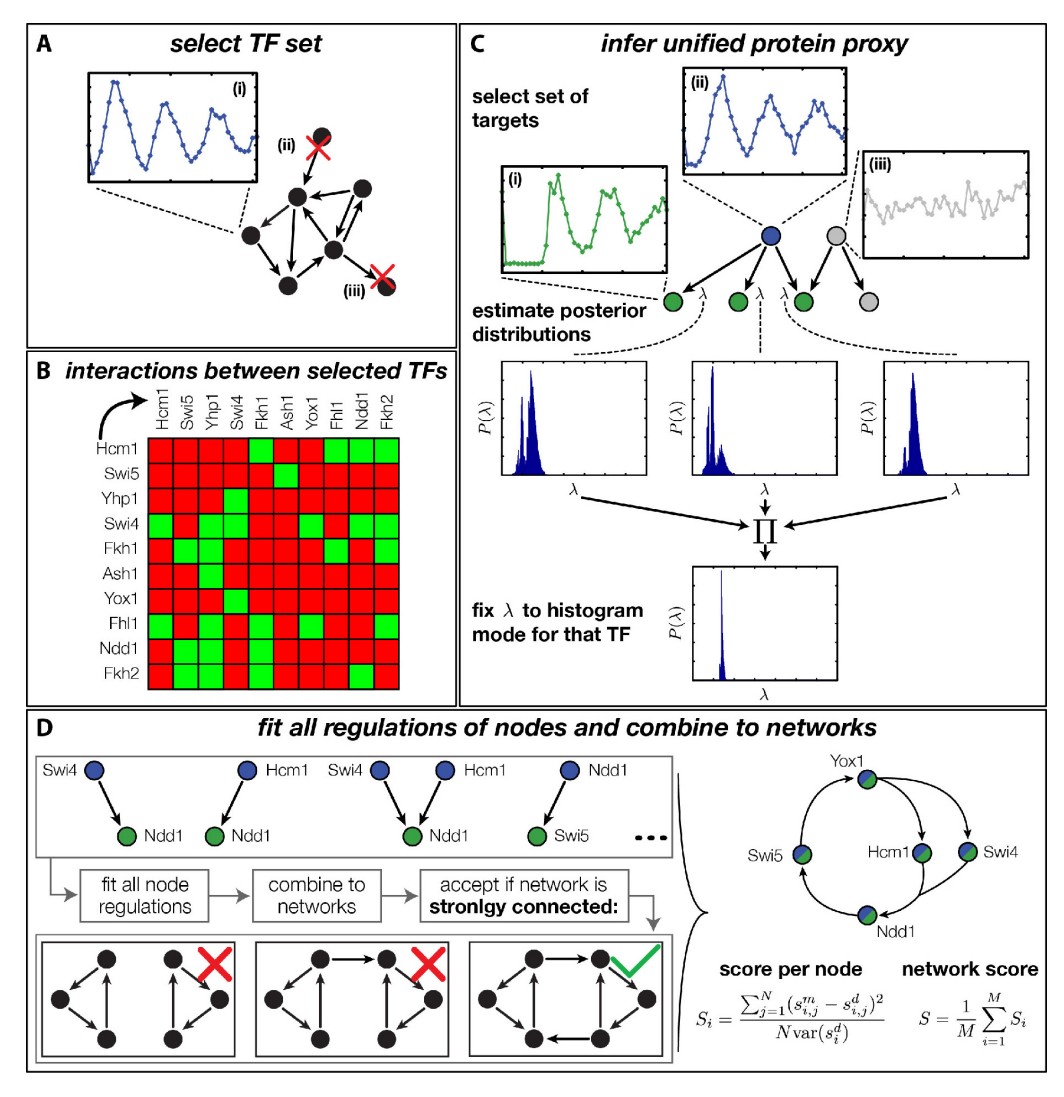

**Figure 4.** Finding a regulatory model for the transcriptional cell cycle oscillator. (**A**) A set of potentially constituent TFs is selected. We require that these genes are (i) periodically expressed, (ii) regulate at least one other gene within the set and (iii) are regulated by at least one other gene within the set. (**B**) Interaction table of the resulting set of genes (listed on the axis), as annotated in the literature. A green square indicates that the corresponding gene of that row regulates the respective gene on the column. (**C**) Before fitting regulation functions to target genes we pre-determine the effective protein degradation rate for each TF. To that end we first select a set of target genes, which are (i) periodically expressed, (ii) are well fitted by the TF and (iii) if there are other TFs regulating the target gene, they must be non-periodic so that they cannot contribute in explaining the expression pattern of the target gene. We then use MCMC to sample from the posterior distribution of the fitting parameters of each target gene and create histograms for the marginal distributions over the protein parameter. The consensus is fixed to the highest peak of the product histogram. (**D**) To each gene in the set of potentially constituent gene regulation models are fitted for all single inputs and all combinations of two inputs. To each fit we calculate a normalized score as the averaged squared residuals of the fit, divided by the data variance of the target synthesis rate. These 'regulatory nodes' are combined to networks, which we require to be strongly connected (as illustrated on the bottom). The resulting set of possible network models are scored by the average normalized fitting score of the nodes and ranked.

The following source data and figure supplements are available for figure 4:

**Source data 1.** Table of interactions between cell cycle TFs with references.
**Source data 2.** Data for protein model inference.
**Source data 3.** List of inferred regulation directions of genes in the trancriptional cell cycle oscillator
**Figure supplement 1.** Fitted GRF for *Hcm1*.

*Figure 4 continued on next page*

*Figure 4 continued*

**Figure supplement 2.** Fitted GRF for *Yhp1*.

**Figure supplement 3.** Fitted GRF for *Swi4*.

**Figure supplement 4.** Fitted GRF for *Fkh2*.

**Figure supplement 5.** Fitted GRF for *Fkh1*.

**Figure supplement 6.** Fitted GRF for *Yox1*.

**Figure supplement 7.** Fitted GRF for *Fhl1*.

**Figure supplement 8.** Fitted GRF for *Ndd1*.

study. Since Swi5 already has all the inputs that Ace2 has within our set, the combined node is simply referred to as Swi5 in the following.

*Figure 4B* shows the interaction matrix for the remaining set of 10 candidate network nodes. This matrix includes the direct interactions from the YEASTRACT database, which are experimentally confirmed TF-TF interactions (see *Figure 4—source data 1* for the original reference for each interaction). Additionally, we included regulation of the *Swi4* gene by Yox1 and Yhp1, which can bind to its ECB promoter element (*Mai and Miles, 2002*; *Pramila et al., 2002*; *Darieva et al., 2010*).

We next estimated the effective protein degradation rate $\lambda$ for all TFs within our set. We exploited the fact that each of these TFs also has multiple target genes outside of the set. Since the parameter $\lambda$ is a property of the TF and not of its targets, the value of $\lambda$ should be consistent between all targets. We therefore devised a method that infers a unified proxy for $\lambda$ from a set of target genes. The method is illustrated in *Figure 4C* and detailed in 'Materials and methods'. Briefly, for each TF in the set we selected target genes, which have a significant regulatory input from the TF over the cell cycle. We then used Markov Chain Monte Carlo sampling to estimate posterior distributions for the parameter $\lambda$ at each target gene, and combined the results into a unified estimate (*Figure 4—source data 2*). The corresponding effective protein half-lives $\tau_{1/2} = \ln(2)/\lambda$ range from 5 min (Ash1) to 50 min (Yox1) and provide the basis for our unified proxy $p(t)$ for the protein dynamics of each TF.

In the crucial final step, we constructed multiple candidate GRFs for each gene in our set and combined them into a large number of candidate network models (*Figure 4D*). For each gene, we fitted GRFs with all possible regulations involving either one or two inputs chosen from the interaction matrix of *Figure 4B*. Each fit has an associated score normalized such that it measures the fraction of data variance not described by the fitted model (see 'Materials and methods'). This assures a meaningful relative weighting of the individual nodes in the combined network scores used below. Two examples for GRFs of best-scoring regulations are shown in *Figure 5*, the regulation of *Ash1* by Swi5 and the regulation of *Swi5* by the two inputs Fkh1 and Ndd1; see *Figure 4—figure supplements 1–8* for the remaining best-scoring GRFs. Importantly, all best-scoring GRFs correctly predicted whether a TF is activating or repressing, wherever experimental evidence exists (see *Figure 4—source data 3*; *Chua et al., 2006*; *Di Talia et al., 2009*). Furthermore, the shape of the GRFs obtained from the two biological replicates in the DTA data set were reasonably consistent, as also illustrated by the examples in *Figure 5*.

In principle, 116,872,448 different gene networks can be combinatorially constructed from our 10 nodes using the different regulations that we allow for each node. However, out of these only 952,100 satisfy our criterion of being strongly connected. Their score distribution is shown in *Figure 5—figure supplement 1A*. For the construction of candidate networks, we do not only allow the best-scoring regulations, but also the suboptimal ones, since due to the constraints the globally optimal network does not necessarily consist only of optimal nodes. The globally optimal genetic network depicted in *Figure 5A* has a strong overall similarity to transcriptional oscillator networks that

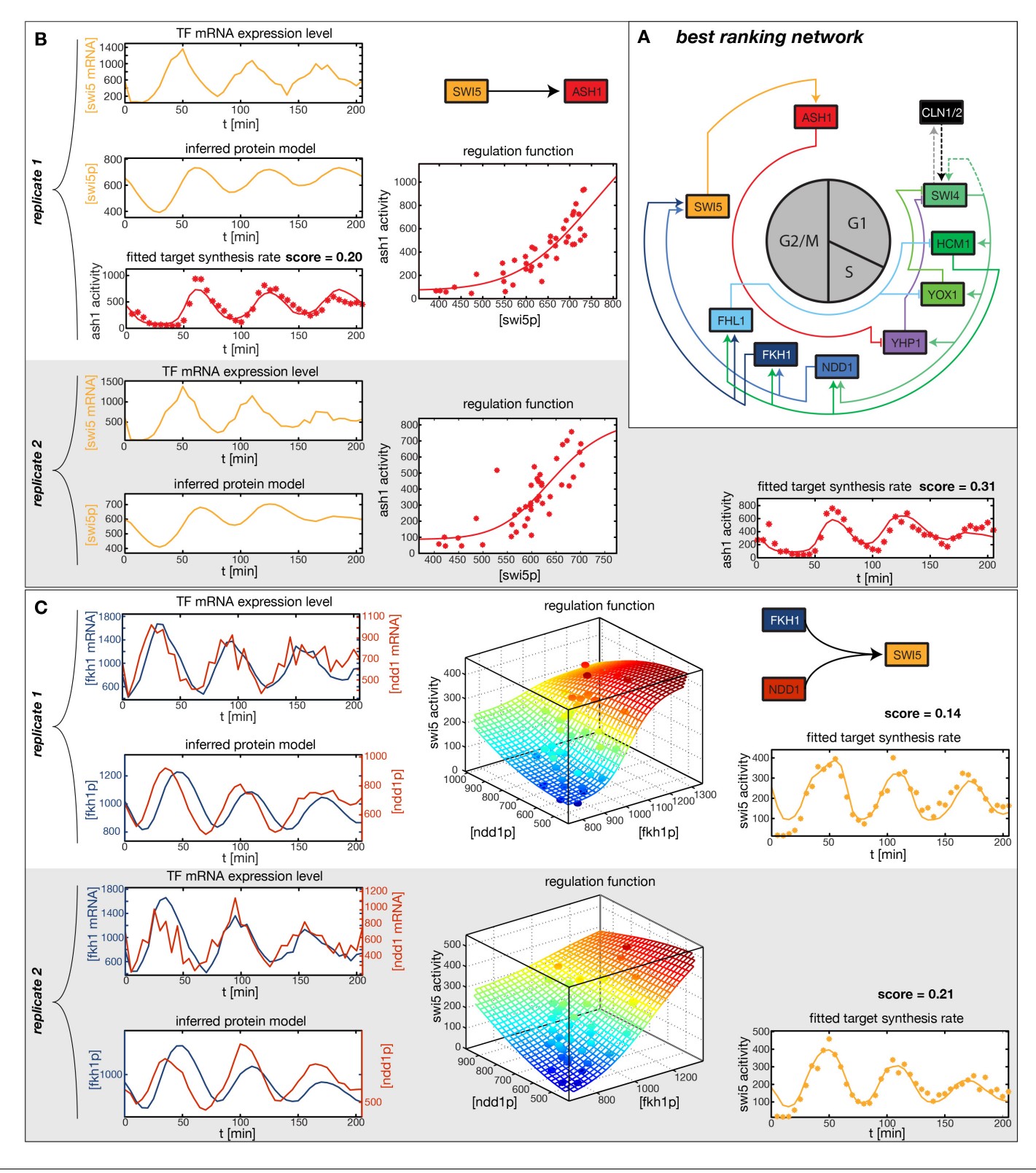

**Figure 5.** Results of node fitting in the cell cycle network. (**A**) Regulation diagram of our best ranking model for a transcriptional cell cycle oscillator. Arrows represent positive (pointing clockwise) and negative (pointing counter-clockwise) interactions and are color-coded with respect to their corresponding TF. Only the regulatory interactions indicated by solid lines resulted from our network inference method, while the additional interactions indicated by dashed lines were introduced in the subsequent manual analysis (see main text). (**B**) Example of a regulation model with two

*Figure 5 continued on next page*

*Figure 5 continued*

input factors: *Swi5* is regulated by Fkh1 and Ndd1 by an analog XOR-like function. (**C**) Example of a one input factor regulation model: *Ash1* is activated by Swi5. Depicted in (**B**) and (**C**) are, for both replicates respectively, data of the TF(s) total mRNA expression level, the inferred TF protein time series, the fitted regulation function and the resulting target synthesis rate. Data points of the target synthesis rate are displayed as stars or colored spheres.

The following figure supplements are available for figure 5:

**Figure supplement 1.** Network scores and best ranking networks.

**Figure supplement 2.** The GRF for *Swi4* is an outlier.

**Figure supplement 3.** Current biological model and the effective model for regulation of *Swi4*.

**Figure supplement 4.** GRF for *Swi4* with cyclin input.

**Figure supplement 5.** Concerted regulation by both Swi4 and Ash1 are necessary to model the expression pattern of *Yhp1*.

**Figure supplement 6.** Concerted regulation by both Hcm1 and Ndd1 are necessary to model the expression pattern of *Fkh1*.

---

were previously proposed. It contains all of our candidate nodes except *Fkh2*. All nodes with the exception of *Ash1* have two regulatory inputs. To confirm that multiple inputs are indeed necessary to explain the expression pattern of these genes, we compared the best fits with inputs from only a single TF with the respective combinatorial regulation. *Figure 5—figure supplement 5* illustrates this for the case of *Yhp1* regulation by Swi4 and Ash1, while *Figure 5—figuer supplement 6* shows analogous plots for *Fkh1* regulation by Hcm1 and Ndd1.

*Figure 5—figure supplement 1* compares our five best-scoring networks. The networks on rank 2, 3, and 5 have the same nodes as our best-ranking network. They differ only by the absence of one regulatory input at a single gene, supporting the best-ranking network as a consensus. The network on rank 4 features *Fkh2* as additional node, and differs in several regulatory interactions, making it incompatible with our consensus network. Taken together, these results illustrate that our inference method extracts useful regulatory information about genetic modules from DTA data. However, we will see in the next section that further manual analysis of the output generated by the inference method can considerably increase the biological insight.

## Analysis of the transcriptional cell cycle oscillator network

By design the gene regulatory network obtained in the previous section does not include regulation by the 'external' cyclin-CDK oscillator. However, further analysis suggests that a completely autonomous network is incompatible with our data. Close inspection of the fitting results reveals that the expression dynamics of the *Swi4* gene is not adequately explained even by the best-scoring GRF. Indeed, the *Swi4* node already stands out in the score statistics (*Figure 5—figure supplement 2*) – whereas the best-scoring GRFs for the other nodes leave only between 14% and 38% of the variation in the data unexplained, 70% and 52% unexplained variance remain for *Swi4* in replicates 1 and 2, respectively. The best-scoring regulation for *Swi4* is by Yhp1 and Yox1, consistent with the known *Swi4* regulation by Yhp1 and Yox1 during the G1 phase (*McInerny et al., 1997*; *Pramila et al., 2002*). However, the clear difference between the shape of the output profile of *Swi4* and its best fit by the Yhp1 and Yox1 inputs (*Figure 4—figure supplement 3*) suggests that additional regulatory inputs are significantly affecting the transcription rate of the *Swi4* gene.

The molecular role of Swi4 in cell cycle regulation is well studied. Swi4 is the DNA binding component of the SBF complex, which activates *Cln1/Cln2* (*Koch et al., 1996*) and late G1 genes, thereby promoting progression from G1 to S phase (*Nasmyth and Dirick, 1991*). Further, Swi4 activates its own expression via SBF (*Iyer et al., 2001*). SBF activity is inhibited in early G1 by the repressor Whi5 (*de Bruin et al., 2004*; *Costanzo et al., 2004*), which in turn is exported from the nucleus at cell cycle START, promoted by G1 cyclins Cln1-3 and Cdc28 (*Wijnen et al., 2002*; *de Bruin et al., 2004*; *Costanzo et al., 2004*). Commitment to START is primarily achieved by a positive feedback

loop between SBF and Cln1/Cln2 to rapidly exclude the SBF repressor Whi5 from the nucleus (*Bean et al., 2006*; *Skotheim et al., 2008*). For the purpose of our study, we modeled this complex set of interactions by a reduced regulation scheme, as illustrated in *Figure 5—figure supplement 3*. Following *Skotheim et al. (2008)*, we considered Cln1/2 to be the main control of Whi5 nuclear export. This led us to an effective scheme where *Swi4* has Cln1/2 and Swi4 itself as additional inputs (indicated by the dashed links in *Figure 5A*). With these added inputs and a matching GRF (see 'Materials and methods'), the *Swi4* output profile was equally well described as the other nodes of the network (*Figure 5—figure supplement 4*).

So far we considered the dynamics at each node separately from the other nodes, and it remains to be shown that our entire network model as a dynamical system generates oscillations compatible with the dynamic transcriptome data. We therefore used the inferred GRFs and effective protein half-lives to construct a complete set of differential equations for the mRNA expression levels and protein concentrations in our network, see 'Materials and methods'. We integrated these equations to simulate the dynamics of the full network starting from the measured mRNA levels and inferred protein concentrations at $t = 0$ as initial values. In addition, we fed the measured dynamics of Cln2 as an external input into our network acting on the *Swi4* node.

*Figure 6* shows the obtained model trajectories for the mRNA expression levels of all nodes, together with the corresponding experimental data. For both replicates, the model adequately captures the behavior of the dynamic transcriptome data. This agreement prompted us to test if our simulation model could also qualitatively predict the behavior of mutant strains. Towards this end, we considered expression data from two studies with TF knockout strains (*Bean et al., 2005*; *Pramila et al., 2002*) and compared it to the behavior of our simulation model with the expression levels of the mutated genes set to zero.

*Bean et al. (2005)* measured the effects of deleting *Swi4* using the *cdc20* block-release protocol for cell-cycle synchronization. The wild-type mRNA expression time series plotted in *Figure 1*, of *Bean et al. (2005)* qualitatively resembles the corresponding data of *Eser et al. (2014)* after taking into account an apparent time-shift, which is likely due to the different protocol for cell-cycle synchronization. In their *Swi4* deletion strain, *Bean et al. (2005)* found a strongly reduced expression of *Yox1* with a weak rapid oscillation (period $\sim 45$ min). This phenotype is recapitulated by our model prediction, see *Figure 6—figure supplement 1A*. In our simulation, the rapid oscillations are caused by indirect effects via Fhl1, the second input of *Yox1*.

A second case is shown in panel B of the same figure, where the expression of the *Swi4* gene is considered in a $\Delta yox1\Delta yhp1$ double mutant background, as studied experimentally by *Pramila et al. (2002)*. The microarray data exhibits a delayed and prolonged peak expression, which reaches into S and G2 phase (*Pramila et al., 2002*). Our model qualitatively reproduces this behavior when *Yox1* and *Yhp1* expression is set to zero. It should be noted, however, that the effect is significantly more pronounced in the model than in the data, suggesting that our model for *S. cerevisiae* transcriptional cell cycle oscillations is missing a mechanism that buffers against the effects of the $\Delta yox1\Delta yhp1$ double mutation.

While the qualitative agreement obtained in *Figure 6—figure supplement 1A and B* suggests that our inferred model indeed captures important aspects of the transcriptional regulation of cell cycle genes, it is clear that it will fail to predict the effect of deletions that unmask post-transcriptional effects. This is illustrated in *Figure 6—figure supplement 1C* for the case of *Rnr1*, a target gene of the transcription factors Swi4 and Mbp1. While the periodic expression of *Swi4* can explain the cell cycle dependent activity of the SBF complex (Swi4-Swi6), *Mbp1* is not periodically transcribed, and the activity of the MBF complex (Mbp1-Swi6) is likely cell cycle dependently modulated by cyclin-dependent posttranscriptional regulation (*de Bruin et al., 2008*). Accordingly, our method infers a GRF based almost entirely on regulatory input from Swi4, and predicts that the transcription rate of *Rnr1* is essentially constant in a *Swi4* mutant strain, as shown in *Figure 6—figure supplement 1C*. In contrast, the *Swi4* deletion strain of *Bean et al. (2005)* exhibits an oscillatory expression of *Rnr1*, with a delayed peak time and an increased amplitude. As shown by *Bean et al. (2005)*, *Rnr1* is in fact redundantly regulated by SBF and MBF, such that only the swi4-mbp1 double mutant displays an *Rnr1* expression that is essentially cell-cycle-independent.

Finally, we returned to the global behavior of the transcriptional oscillator. Given that Cln2 already provides an oscillatory input signal to our simulation model, we can conclude only that the network can be driven towards transcriptional oscillations with the correct amplitudes, shapes, and

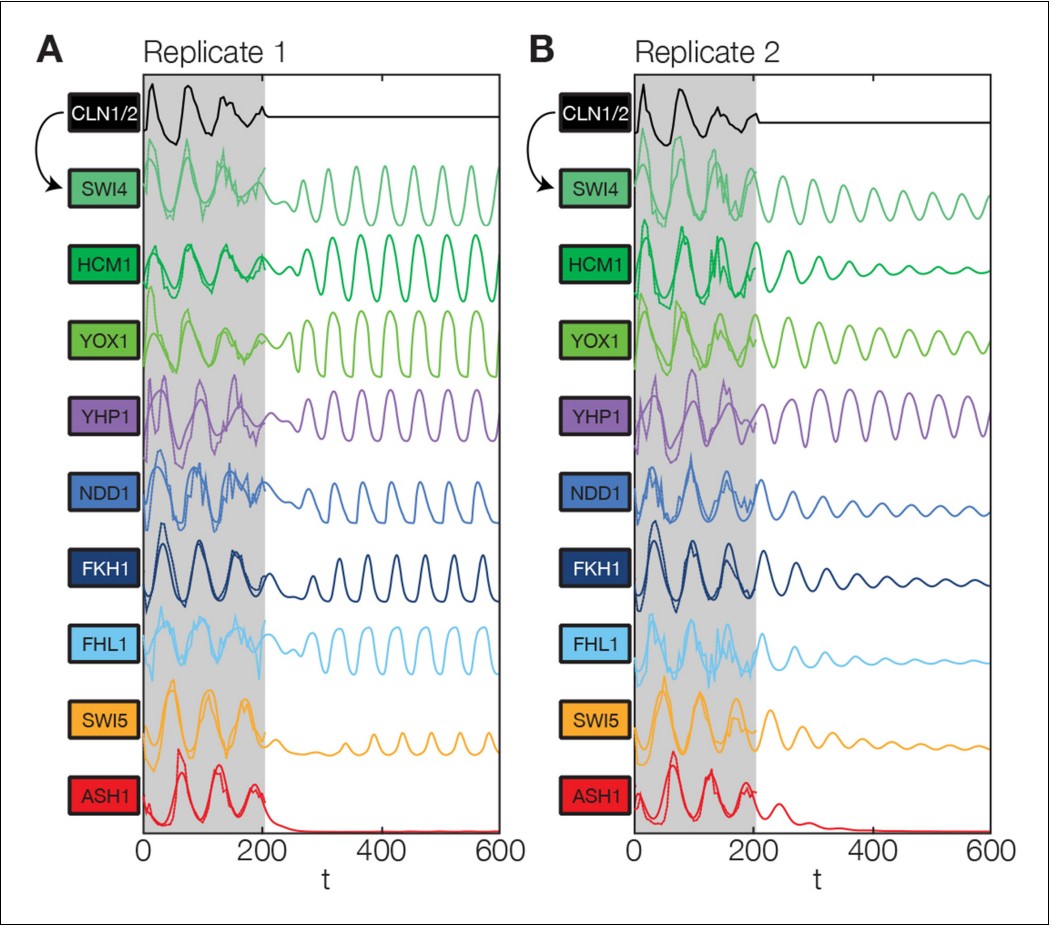

**Figure 6.** Simulation results of the reconstructed transcriptional cell cycle oscillator. Simulated mRNA expression levels for all genes in the network are shown in arbitrary units and rescaled to equal sizes. For the first 205 min (shaded grey) *Swi4* receives periodic input from Cln2 using measured expression data. After 205 min the input from Cln2 is set to a constant average value to simulate loss of cyclin activity. (**A**) The network inferred from replicate dataset 1 recovers sustained oscillations. (**B**) Oscillations in the network inferred from replicate dataset 2 dampen over time without periodic input.

The following figure supplements are available for figure 6:

**Figure supplement 1.** Test of predictions for mutant strains by model simulations.

**Figure supplement 2.** Relative change of model parameter after re-fitting the full ODE model compared to the individual fits.

relative phases in all of its nodes. Can it also oscillate autonomously, without external driving? To test this, we continued to run the network dynamics simulation beyond the last measured timepoint (t=205 min) with the Cln2 input set to zero, mimicking the loss of cell cycle dependent cyclin activity.

We found that the network has the intrinsic ability to oscillate. After an initial disturbance due to the abrupt disappearance of the Cln2 input, regular oscillations were recovered in most genes for replicate 1 (*Figure 6A*), while dampened oscillations were obtained for replicate 2 (*Figure 6B*). Hence, depending on the parameter values, which were separately adjusted to describe each data set (see 'Materials and methods'), the network model is able to produce either regular or dampened oscillations. However, one node of the network appears to require the Cln2 input for oscillation: for both replicates Ash1 oscillations cease almost immediately after the Cln2 input is taken away. Interestingly, Ash1 has also been reported to be non-oscillating in some cyclin mutant strains

(*Orlando et al., 2008*), suggesting that its ability to oscillate is particularly sensitive with respect to perturbations of the cyclin-CDK oscillator.

While most genes still oscillate, the oscillation period is shortened by $\sim 20\%$ in the absence of the Cln2 input, underlining the importance of cyclin-CDK activity for the timing of gene expression. Despite this change in the absolute period, the relative ordering of the genes in terms of the phase of their oscillations remains the same as with the Cln2 input. Taken together, we arrived at a model for a transcriptional cell cycle oscillator that (i) is quantitatively consistent with the dynamic transcriptome data for synchronized wild-type cells, and (ii) can produce (dampened) oscillations without cyclin input, albeit not with the native properties, supporting the notion that cyclin action is required for normal oscillatory behavior (*Simmons Kovacs et al., 2012*; *Rahi et al., 2016*).

## Discussion

We established a method to infer gene regulation functions (GRFs) from the intrinsic cellular dynamics of the transcriptome. GRFs are key quantities for the modeling of transcription regulation and provide a basis for the quantitative analysis of functional genetic modules. Inference of GRFs is necessary, since their direct measurement is notoriously difficult while indirect measurements are limited to specific TFs (*Setty et al., 2003*) and confounded by extraneous factors (*Kuhlman et al., 2007*). We demonstrated, for the first time, the inference of individual GRFs as well as a functional genetic module from dynamic transcriptome data, which in our example was obtained from wild-type synchronized yeast cells.

Our inferred GRFs agreed well between biological replicates, were able to capture the expression dynamics of the target genes also in an independent test dataset, and correctly predicted whether the effect of a TF on its target is activating or repressing wherever experimental evidence was available. The method identifies the two inputs of a gene that are most significant for the description of the experimental data set. We illustrated the consistency of this reduced description with a more detailed physico-chemical model that takes all known inputs into account. The amount of data needed to infer GRFs rises strongly (exponentially) with the number of regulatory inputs, due to the combinatorial complexity of the task. Given the data that is currently available, our inference of GRFs with two inputs is at the limit of what is possible in a systematic and unbiased way.

Clearly, GRFs could be extracted more directly and reliably if the time-dependent protein levels of all input TFs were also measured, as well as their activity and nuclear localization. However, we showed that one can obtain surprisingly consistent GRFs from high-accuracy dynamic transcriptome data alone, which simultaneously provides the transcript levels of the inputs and the mRNA synthesis rates of their targets. These GRFs are not based on detailed mechanistic models of the underlying molecular processes, but correspond to effective regulation models which subsume many processes, including the transport of the TFs to and from the nucleus and possibly activation of the TFs, e.g. by phosphorylation.

The ability of our method to infer GRFs from dynamic transcriptome data depends on two general conditions: First, prior knowledge about which TFs potentially regulate which targets is required, since the information contained in the time series does not suffice to discriminate the correct regulatory interactions from all possible wrong associations. For the examples that we considered, the prior information provided by the YEASTRACT database proved sufficient. This database lists more input TFs than are relevant under the conditions that we study. However, since the number of irrelevant interactions is small (compared to the number of all possible inputs), our method is able to identify the most relevant ones. Second, the information contained in the data must be sufficiently redundant to permit self-consistency conditions to be imposed during the inference procedure. We made use of redundant information derived from the length of the time series (the transcriptome data covers three cell cycles) and the pleiotropy of the cell cycle transcription factors (which must produce consistent effects in multiple targets). Self-consistency conditions are essential for our method to circumvent the need for protein data.

One of the most promising applications of GRF inference is the quantitative analysis of functional genetic modules. Modules composed of interacting molecules and genes are widely considered central for the understanding of cellular functions (*Hartwell et al., 1999*). While the topologies of possible genetic modules can be generated from known regulatory interactions, any attempt to quantitatively test the compatibility of such modules with expression data will have to infer the GRFs

of the modules nodes. We introduced and illustrated an unbiased approach to this task. Instead of testing a small set of preselected candidate network modules against the dynamic transcriptome data, we systematically assessed close to a million candidates for the core module of the yeast transcriptional cell cycle network.

Our best ranking network has several features in common with previously proposed transcriptional cell cycle networks (*Simon et al., 2001*; *Lee et al., 2002*; *Orlando et al., 2008*; *Simmons Kovacs et al., 2012*). Compared to the latest proposal, which was constructed by hand from genes that remained periodic in a *cdc28* mutant (*Simmons Kovacs et al., 2012*), our best ranking network contains the same genes, except that it adds the nodes Yox1 and Fkh1 but does not explicitly consider Mcm1 as significant regulatory input (although Mcm1 certainly plays an important mechanistic role). Our overall network topology is also similar, with sequential forward activation and backward inhibition as dominant motifs, consistent with previous studies (*Simon et al., 2001*; *Orlando et al., 2008*; *Simmons Kovacs et al., 2012*). Beyond recapitulating prior results, our unbiased inference method has led us to a mathematical model that is consistent with the DTA data and capable of generating oscillations, while also predicting a coupling to the cyclin-CDK oscillator.

The mathematical models obtained with our inference approach are valuable for further quantitative analysis. As an illustration, we tested to what extent our core module could generate oscillations without input from the cyclin-CDK oscillator. While its dynamics was sensitive to loss of the cyclin-CDK input, it was still able to display either regular or dampened oscillations. This behavior is reminiscent of the diverse dynamical behavior generated by synthetic biochemical oscillators for different initial conditions or parameters (*Weitz et al., 2014*). Our results support a picture whereby the transcriptional oscillations do not strictly depend on the cyclin input, but are intimately coupled with the post-translational oscillations to produce the observed wild-type behavior (*Simmons Kovacs et al., 2012*). While our model cannot resolve the detailed mechanism of the coupling, e.g. because it cannot take interactions with cell cycle checkpoints into account (*Bristow et al., 2014*), it does point towards an interesting possibility to reconcile the seemingly contradictory experimental results of *Simmons Kovacs et al. (2012)* and *Rahi et al. (2016)*: The model results of *Figure 6* indicate that the transcriptional oscillations can change from continuous to dampened within a narrow parameter range. *Simmons Kovacs et al. (2012)* measured periodic transcription immediately after disabling a temperature sensitive allele of *cdc28* by a shift to the restrictive temperature. In contrast, *Rahi et al. (2016)* applied a 'cyclin depletion protocol' to clear all cyclins from the cells before testing for transcriptional oscillations. Dampened oscillations might indeed account for both observations, a periodic transcription with a decaying amplitude immediately after loss of G1 cyclin activity, and cessation of all periodic events after an extended 'cyclin depletion protocol'.

The present study provides a proof of principle for our approach for extracting quantitative information about GRFs from gene expression time series. Other studies have already demonstrated the rich information that dynamic expression data provide about regulatory interactions (*Dunlop et al., 2008*; *Lipinski-Kruszka et al., 2015*) and regulatory mechanisms (*Westermark and Herzel, 2013*). In the future, we expect that inference and quantitative analysis of functional modules will develop into an extremely powerful approach as dynamic transcriptome data is systematically collected for genetic variants and under different physiological conditions. With suitable data, the approach could be extended to include inputs from regulatory RNAs and regulation on the post-transcriptional level.

## Materials and methods

### Experimental data acquisition

The dataset used in this work (cDTA for the yeast cell cycle) has been previously published (*Eser et al., 2014*). The complete dataset is available at ArrayExpress (RRID:SCR_002964) under accession code E-MTAB-1908.

### Model and parameter optimization for gene regulation functions

To fit GRFs to the time series of input and output signals, we use a score function that quantifies the fraction of the variance in the output signal that is not explained by the candidate GRF with the given input signals. For a given target gene $i$, the score $S_i$ is defined as a ratio, where the numerator corresponds to the mean squared deviation between the genes measured synthesis rate $s_i^d(t_n)$ and

the corresponding model synthesis rate $s_i^m(t_n)$, averaged over all 42 time points in the dataset ($t_n = 0, 5, 10, \ldots, 205$ min). The denominator corresponds to the variance of the time series $s_i^d(t_n)$, such that

$$S_i = \frac{1}{N}\sum_{n=1}^{42}\left(s_i^d(t_n) - s_i^m(t_n)\right)^2 \bigg/ \frac{1}{N}\sum_{n=1}^{42}\left(s_i^d(t_n) - \langle s_i^d \rangle\right)^2$$

where $\langle s_i^d \rangle$ denotes the time-averaged measured synthesis rate of gene $i$. The score $S_i$ is 0 for a perfect fit and 1 if the model is just a constant value corresponding to the average of the expression data. The fitting task is to minimize the score with respect to the model parameters. The model consists of two parts, the model for the protein dynamics and the parameterization of the different types of GRFs.

The GRFs are functions of the protein levels of the input TFs. We model the dynamics $p_i(t)$ of each TF protein level by

$$\frac{dp_i(t)}{dt} = \nu_i m_i(t) - \lambda_i p_i(t),$$

with a translation rate $\nu_i$ and an effective degradation rate $\lambda_i$, as discussed in the main text. This generates a time series for the TF protein level,

$$p_i(t_n) = p_i(0)e^{-\lambda_i t_n} + e^{-\lambda_i t_n}\int_0^{t_n}\nu_i e^{\lambda_i \tau}m_i(\tau)\,d\tau,$$

where the mRNA time series $m_i(t_n)$ is interpolated by cubic splines and the integral is evaluated numerically. The initial values $p_i(0)$ are fixed by first solving the equation until the initial condition has essentially decayed and then extrapolating backwards to obtain a smooth expression pattern. The ratio $\nu_i/\lambda_i$ sets the absolute level of protein relative to the mRNA level, while the dynamics of the relative protein level is governed by the parameter $\lambda_i$ alone. The absolute protein levels are not relevant for our analysis, since all GRFs respond to ratios of protein levels to effective binding constants that are on an arbitrary scale here. Thus, we can set the ratio $\nu_i/\lambda_i$ to one, such that $dp_i/dt = \lambda_i(m_i - p_i)$. This leaves $\lambda_i$ as the only protein model parameter to be inferred for each TF.

Our GRFs are parameterized via Hill functions as shown in *Figure 1* and listed below. We consider two different forms of one-input GRFs, corresponding to activation and repression, and ten different forms of two-input GRFs, corresponding to analog versions of the standard Boolean logic functions. For the one-input GRFs, the parameters are the basal transcription rate $b$, the maximal fold change $\alpha$, the sensitivity of the response $n$, and the TF concentration $K$ at which the target gene is half activated or repressed, respectively. Two-input GRFs involve a $K$ and $n$ parameter for each input and a global $b$ and $\alpha$ parameter. We consider the following types of two-input GRFs:

$$p_1 \,\mathrm{AND}\, p_2: \quad s^m = b + \alpha\left[\frac{p_1^{n_1}}{K_1^{n_1}+p_1^{n_1}} \times \frac{p_2^{n_2}}{K_2^{n_2}+p_2^{n_2}}\right]$$

$$p_1 \,\mathrm{OR}\, p_2: \quad s^m = b + \alpha\left[\frac{p_1^{n_1}}{K_1^{n_1}+p_1^{n_1}} + \frac{p_2^{n_2}}{K_2^{n_2}+p_2^{n_2}}\right]$$

$$p_1 \,\mathrm{AND\,NOT}\, p_2: \quad s^m = b + \alpha\left[\frac{p_1^{n_1}}{K_1^{n_1}+p_1^{n_1}} \times \frac{K_2^{n_2}}{K_2^{n_2}+p_2^{n_2}}\right]$$

$$\mathrm{NOT}\, p_1 \,\mathrm{AND}\, p_2: \quad s^m = b + \alpha\left[\frac{K_1^{n_1}}{K_1^{n_1}+p_1^{n_1}} \times \frac{p_2^{n_2}}{K_2^{n_2}+p_2^{n_2}}\right]$$

$$p_1 \,\mathrm{OR\,NOT}\, p_2: \quad s^m = b + \alpha\left[\frac{p_1^{n_1}}{K_1^{n_1}+p_1^{n_1}} + \frac{K_2^{n_2}}{K_2^{n_2}+p_2^{n_2}}\right]$$

$$\mathrm{NOT}\, p_1 \,\mathrm{OR}\, p_2: \quad s^m = b + \alpha\left[\frac{K_1^{n_1}}{K_1^{n_1}+p_1^{n_1}} + \frac{p_2^{n_2}}{K_2^{n_2}+p_2^{n_2}}\right]$$

$$p_1 \,\mathrm{NOR}\, p_2: \quad s^m = b + \alpha\left[\frac{K_1^{n_1}}{K_1^{n_1}+p_1^{n_1}} \times \frac{K_2^{n_2}}{K_2^{n_2}+p_2^{n_2}}\right]$$

$$p_1 \,\mathrm{NAND}\, p_2: \quad s^m = b + \alpha\left[\frac{K_1^{n_1}}{K_1^{n_1}+p_1^{n_1}} + \frac{K_2^{n_2}}{K_2^{n_2}+p_2^{n_2}}\right]$$

$$p_1 \,\mathrm{XOR}\, p_2: \quad s^m = b + \alpha\left[\left(\frac{p_1^{n_1}}{K_1^{n_1}+p_1^{n_1}} \times \frac{K_2^{n_2}}{K_2^{n_2}+p_2^{n_2}}\right) + \left(\frac{K_1^{n_1}}{K_1^{n_1}+p_1^{n_1}} \times \frac{p_2^{n_2}}{K_2^{n_2}+p_2^{n_2}}\right)\right]$$

$$p_1 \,\mathrm{EQ}\, p_2: \quad s^m = b + \alpha\left[\left(\frac{p_1^{n_1}}{K_1^{n_1}+p_1^{n_1}} \times \frac{p_2^{n_2}}{K_2^{n_2}+p_2^{n_2}}\right) + \left(\frac{K_1^{n_1}}{K_1^{n_1}+p_1^{n_1}} \times \frac{K_2^{n_2}}{K_2^{n_2}+p_2^{n_2}}\right)\right]$$

To minimize the score and identify the best-fit GRF, we use simulated annealing with a self-adapting cooling schedule (*Lam and Delosme, 1988*) to find the global minimum of the score as a function of

the protein and GRF parameters. We perform this separately for each of the GRF types that we consider and select the one that yields the best score.

## Prediction of binding sites

To predict binding sites in the promoter region of target genes we retrieved 700 bp sequences upstream of the consensus TSS from YEASTRACT (RRID:SCR_006076, *Teixeira et al., 2006*) and matched binding motifs of relevant transcription factors in the JASPAR database (RRID:SCR_003030, *Sandelin et al., 2004*) with a relative score cutoff of 0.8. Additionally, we matched the published consensus regulatory motif of the *CLB2* cluster (*Spellman et al., 1998*), using MAST (RRID:SCR_001783, *Bailey et al., 1998*).

## GRF validation against independent data

Microarray data from *Pramila et al. (2006)* were obtained from the GEO database (RRID:SCR_005012). Because the data are logarithmic, and each gene is individually normalized, we exponentiated the data and performed a linear transformation to make its range comparable to our dataset. The linear transformation, $\tilde{m}_i^t(t) = a_i + b_i m_i^t(t)$ with $m_i^t(t)$ denoting the expression time course of gene $i$ in the test dataset, was performed such that the minimum and maximum expression of each gene in the test data is equivalent to the minimum and maximum in our dataset, i.e., the constants were chosen as $b_i = (\max[m_i] - \min[m_i])/(\max[m_i^t] - \min[m_i^t])$ and $a_i = \min[m_i] - b_i \min[m_i^t]$. The obtained expression time series for the TFs, $\tilde{m}_i^t(t)$, was then used together with the GRFs inferred from our data to generate predictions for the target gene expression to be compared with the corresponding values in the *Pramila et al. (2006)* dataset.

## Inference of multi-gene networks

Documented regulatory interactions between TFs and target genes have been downloaded from the YEASTRACT database (http://yeastract.com/download/, *Teixeira et al., 2006*) and matched to the set of genes measured in the DTA experiments. Based on this information, TFs satisfying conditions (ii) and (iii) in *Figure 4A* were selected. To estimate a unified protein proxy for a TF we first select a set of suitable target genes by the following requirements: (i) the target genes must be periodically expressed (analogously to requirement (i) in *Figure 4A*); (ii) for each target gene there must be a GRF (single input or combinatorial) with the TF as input and a score lower than 0.5; (iii) each target gene is fitted with a GRF in which the regulatory sign of the TF (activator or repressor) corresponds to the literature reference (c.f. *Figure 4—source data 3*). On each target gene of the resulting set a MCMC algorithm is performed to sample the score function of the corresponding GRF and a histogram of the sampled posterior distribution over the parameter $\lambda$ is generated (*Hastings, 1970*; *Andrieu et al., 2003*). Finally the posterior distributions of all target genes are combined by histogram multiplication and the mode of the product histogram is determined to be the estimated parameter of the unified protein proxy of the TF.

## Regulation model for Swi4

We model coupling to the primary cyclin-CDK oscillator by placing *Swi4* under the control of Cln2 and itself (via SBF). We include regulation by Cln2 and Swi4 as additional (additive) activators:

$$s_{\text{swi4}}^m = b + \alpha \left[ \left( \frac{K_{\text{yhp1}}^{n_1}}{K_{\text{yhp1}}^{n_1} + p_{\text{yhp1}}^{n_1}} \times \frac{K_{\text{yox1}}^{n_2}}{K_{\text{yox1}}^{n_2} + p_{\text{yox1}}^{n_2}} \right) + \left( \frac{p_{\text{cln2}}^{m_1}}{K_{\text{cln2}}^{m_1} + p_{\text{cln2}}^{m_1}} \times \frac{p_{\text{swi4}}^{m_2}}{K_{\text{swi4}}^{m_2} + p_{\text{swi4}}^{m_2}} \right) \right].$$

## Network dynamics

In a network with $N$ interacting transcription factors, we describe mRNA and protein expression level of each gene $i$ by the following equations:

$$\begin{aligned} \frac{dm_i(t)}{dt} &= s_i^m(p_j(t), \ldots) - \delta_i m_i(t), \\ \frac{dp_i(t)}{dt} &= \lambda_i(m_i(t) - p_i(t)). \end{aligned} \tag{1}$$

The mRNA synthesis rate $s_i^m$ is given by the regulation model found by the reconstruction method and coupled, correspondingly, to the protein expression level of other TF's in the model. In this coarse-grained description of a gene regulatory network we assume the mRNA degradation rate $\delta_i$

to be constant (as the protein degradation rate $\lambda_i$). We adjust the parameters found by the reconstruction method by minimizing the squared residues between modeled and measured mRNA expression level, normalized for each gene by the data variance (as for the normalized node score defined in *Figure 3D*). Since all nodes had been fitted individually, errors at each node propagate to downstream nodes in a simulation, distorting their behavior. Therefore, we re-fitted the global output of the ODE model to the expression data, using gradient descent and the parameters from the previous fits as starting values. The relative change in the re-fitted parameters with respect to their original values is shown in *Figure 6—figure supplement 2*. The majority of the parameters change less than 10%. To simulate a TF knockout, we set the expression level of the respective gene to a constant zero throughout the simulation. Since the TF is embedded in an extended network, indirect effects in the expression of target genes can also be captured by this method.

## Acknowledgements

We thank Achim Tresch for useful discussions. UG was supported by the Deutsche Forschungsgemeinschaft (Excellence Cluster NIM) and the Bavarian Research Network for Molecular Biosystems (BioSysNet). PC was supported by the Deutsche Forschungsgemeinschaft (SFB860) and the Volkswagen Foundation.

## Additional information

### Competing interests
PC: Reviewing editor, *eLife*. The other authors declare that no competing interests exist.

### Funding

| Funder | Grant reference number | Author |
|---|---|---|
| Volkswagen Foundation | | Patrick Cramer |
| Deutsche Forschungsgemeinschaft | SFB860 | Patrick Cramer |
| Deutsche Forschungsgemeinschaft | NIM | Ulrich Gerland |
| Bayerisches Staatsministerium für Bildung und Kultus, Wissenschaft und Kunst | | Ulrich Gerland |

The funders had no role in study design, data collection and interpretation, or the decision to submit the work for publication.

### Author contributions
PH, UG, Conception and design, Analysis and interpretation of data, Drafting or revising the article; KCM, Acquisition of data, Analysis and interpretation of data; PC, Conception and design, Drafting or revising the article

### Author ORCIDs
Patrick Cramer, http://orcid.org/0000-0001-5454-7755
Ulrich Gerland, http://orcid.org/0000-0002-0859-6422

## Additional files

### Supplementary files
• Source code 1. Software package for the inference of gene regulation functions MATLAB (RRID: SCR_001622) implementations with complementing C function of all computational methods described in this work are provided in an accompanying source code archive.

## Major datasets

The following previously published dataset was used:

| Author(s) | Year | Dataset title | Dataset URL | Database, license, and accessibility information |
|---|---|---|---|---|
| Eser P, Demel C, Maier K, Schwalb B, Pirkl N, Cramer P, Tresch A | 2014 | Transcription profiling by array of Saccharomyces cerevisiae cells to estimate labeled and total mRNA levels every 5 minutes for three complete cell cycles | https://www.ebi.ac.uk/arrayexpress/experiments/E-MTAB-1908/ | Publicly available at ArrayExpress (accession no: E-MTAB-1908) |

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
