## [Decision Letter]

Thank you for submitting your work entitled "Inference of gene regulation functions from dynamic transcriptome data" for consideration by *eLife*. Your article has been reviewed by two peer reviewers, and the evaluation has been overseen by a Reviewing Editor and Detlef Weigel as the Senior Editor. One of the two reviewers has agreed to reveal his identity: Hernan Garcia.

The reviewers have discussed the reviews with one another and the Reviewing Editor has drafted this decision to help you prepare a revised submission

Summary:

Gene regulatory functions (GRFs) are the fundamental unit of any quantitative description of transcriptional programs. These functions describe the output level of gene expression as a function of the input concentrations of activators and repressors and of the binding site arrangement of these molecules on regulatory DNA. The measurement of these GRFs has been demonstrated repeatedly in the context of synthetic constructs, where DNA regulatory architecture and input transcription factor concentration can be precisely controlled. However, the field is missing technology to go beyond synthetic circuits and systematically expand these dissections to endogenous gene regulatory circuits. Only with the combined ability to assay synthetic and endogenous gene circuits can we develop a predictive understanding of the gene regulatory code underlying cellular decision programs.

In this manuscript, Hillenbrand et al. develop a computational approach to infer GRF from endogenous RNA-Seq data sets. They use mRNA data of oscillatory genes in order to infer the protein concentration of input transcription factors. This inferred protein dynamic is combined with the output mRNA dynamics of target genes in order to obtain GRFs. These GRFs are put in the context of the underlying DNA regulatory architecture using theoretical models based on equilibrium statistical mechanics. Finally, they show how this approach can go beyond the quantification of GRFs to also provide a means to map gene regulatory networks and their quantitative parameters.

Essential revisions:

For acceptance at *eLife*, as rigorous as possible validation of the model is required. This can include checking resulting parameters against published values, as well as against published or novel experimental data confirming any of the parameters, such as through knockout or knockdown or overexpression experiments.

Indeed, an intriguing consequence of plots such as those shown in Figure 3 is that the authors make quantitative predictions about the GRF at input concentration values that are not present in the calibration data set. For example, most 3D plots in the paper extrapolate gene expression beyond the input levels observed in the wild-type. The authors could propose the experiments necessary to test these predictions. Perhaps there are already such datasets publicly available. If not, the authors could generate some to support the model.

---

## [Author Response]

*For acceptance at eLife, as rigorous as possible validation of the model is required. This can include checking resulting parameters against published values, as well as against published or novel experimental data confirming any of the parameters, such as through knockout or knockdown or overexpression experiments.*

*Indeed, an intriguing consequence of plots such as those shown in Figure 3 is that the authors make quantitative predictions about the GRF at input concentration values that are not present in the calibration data set. For example, most 3D plots in the paper extrapolate gene expression beyond the input levels observed in the wild-type. The authors could propose the experiments necessary to test these predictions. Perhaps there are already such datasets publicly available. If not, the authors could generate some to support the model.*

Given that the inferred GRFs are the central output of our method, an as rigorous as possible validation of the GRFs is indeed key to establishing our approach. Our validation rests on multiple independent pieces of evidence. Several of these are based on consistency checks that can be applied on the level of individual GRFs:

1) By construction, our method includes a self-consistency check, which relies on redundancies in the data. Specifically, we used the redundancy created by the length of the time series (the transcriptome data covers three cell cycles) and the pleiotropy of the cell cycle transcription factors (which must produce consistent effects in multiple targets). In our analysis of cell cycle transcription factors, we only accept GRFs which produce a consistent description of the data. Of course, this by itself is not sufficient to assure the validity of a GRF, however it is a useful first test. This criterion also led us to the conclusion that the transcription rate time series of the *swi4* gene is *not* adequately described by regulatory inputs from Yhp1 and Yox1 alone, but has other significant inputs from the cyclin-CDK oscillator (see section “Analysis of the transcriptional cell cycle oscillator network” in the main text).

2) We performed a qualitative test of our inferred GRFs against experimental data, by comparing the shape of the GRFs with the known roles of the input transcription factors as transcriptional activators or repressors. In our analysis of cell cycle transcription factors, we found that all best-scoring GRFs correctly predicted whether the regulatory effect of a transcription factor is activating or repressing, whenever independent experimental evidence was available, see [Supplementary-material SD3-data]. Since the expected sign of the regulatory effect of a transcription factor is not included as prior information in our approach, this constitutes an important qualitative test of our inferred GRFs.

3) Furthermore, transcription factors that form complexes at their target genes are expected to act cooperatively, such that both TFs are needed to exert the regulatory function. Our inference scheme indeed predicted a cooperative AND-like regulation function for target genes of the Clb2 cluster, which are regulated by Fkh2 and Ndd1 in a complex with Mcm1.

4) The fact that the DTA data used for our GRF inference contains two biological replicates allowed us to perform our analysis independently on each replicate. We found that the shapes of the resulting GRFs agree well between these two replicates, as illustrated by the examples in Figure 5.

To go beyond consistency checks for individual GRFs, we tested whether the separately inferred GRFs of mutually regulating genes could be combined into a dynamic model that coherently describes the observed transcription time-series of these genes. We focused on the dynamics of ten transcription factors that display strong cell cycle oscillations and are interconnected into a transcriptional network. A model based on the inferred GRFs was able to capture the synergistic dynamics of these genes. Moreover, it predicted that the oscillatory behavior of the gene *ash1* is particularly sensitive to perturbations of the cyclin-CDK oscillator, consistent with experiments that find *ash1* to be non-oscillating in some cyclin mutant strains. Thereby, we indirectly validated our GRFs by demonstrating their effectiveness to describe the in vivobehavior of a functional genetic module.

Taken together, these tests indicate that the inferred GRFs are adequate coarse-grained descriptions of the complex molecular processes that determine the transcription rate of the associated genes, at least over the range of input concentration values that are present in our calibration data set. As pointed out by the Reviewers, it would be desirable to test whether the inferred GRFs are also valid beyond this range. One way to test this would be to repeat the DTA experiments of Eser et al.(2014) with mutant strains, in which the dynamics of cell cycle transcription factors is significantly perturbed. However, the cost (>100 k€) and required time is prohibitive for such experiments to be done for validation purposes. In response to the comments of the Reviewers, we instead used published data from other groups to probe the ability of our GRFs to predict regulatory effects outside the regime of the calibration data set.

First, we considered microarray data from α-Factor synchronized yeast cells, which follows transcript levels over two cell cycles at 5 min intervals (Pramila et al., 2006). We asked whether the same GRFs that we had previously inferred from the data of Eser et al.(2014) would also be able to describe the mRNA dynamics of this dataset, which we had not used to calibrate our GRFs. The new Figure 3—figure supplement 1 shows mRNA time series from this dataset for 13 genes, which are targets of periodically expressed transcription factors. The superimposed curves show the predictions obtained from our GRFs, using also the same effective protein half-lives that we had previously inferred for the transcription factors (see below for a detailed discussion of the significance and interpretation of these parameters). In each case, the input transcription factors are indicated above the graph. Given that the curves are predictions without fit parameters, the overall agreement is remarkable. The regime of input TF concentrations that is effectively sampled by the data of Pramila et al.(2006) is similar but not identical to the range in our calibration data set, as shown in the new Figure 3—figure supplement 2. The differences are not surprising given that the data of Pramila et al. (2006) was obtained with a different strain in a different lab. Moreover, the observation that our GRFs are able to describe the measured transcriptional output despite some differences in the input regime suggests that our GRFs have predictive power beyond the regime of their calibration.

Second, we considered cell cycle experiments with mutant strains and tested whether a model based on our inferred GRFs is able to recapitulate observed phenotypes (Bean et al., 2005; Pramila et al., 2002). Since the deletion of a regulator can affect the expression of its target gene both directly and indirectly, we used our dynamic model for *S. cerevisiae* transcriptional cell cycle oscillations (Figure 6) as a basis for this test. Within this model, a deletion mutant is mimicked by setting the expression rate of the corresponding gene to zero. For instance, Bean et al.(2005) measured the effects of deleting *swi4* using the *cdc20* block-release protocol for cell-cycle synchronization. The wild-type mRNA expression time series plotted in Figure 1 of Bean et al.(2005) qualitatively resembles the corresponding data of Eser et al.(2014) after taking into account an apparent time-shift, which is likely due to the different protocol for cell-cycle synchronization. Given this difference in the protocol and the fact that the raw data of Bean et al.(2005) is not publicly available, we keep the comparison to our model predictions on a qualitative level. Panel A of the new Figure 6—figure supplement 1 shows the model prediction for the Swi4 target gene *yox1.* In their *swi4* deletion strain, Bean et al.(2005) found a strongly reduced expression of *yox1* with a weak rapid oscillation (period ~45 min). This phenotype is recapitulated by the model prediction in panel A of the new figure.

A second case is shown in panel B of the same figure, where the expression of the *swi4* gene is considered in a *∆yox1∆yhp1* double mutant background, as studied experimentally by Pramila et al. (2002). The microarray data exhibits a delayed and longer peak expression, which reaches into S and G2 phase (Pramila et al., 2002). Our model of the transcriptional cell cycle oscillator qualitatively reproduces this behavior when *yox1* and *yhp1* expression is set to zero. It should be noted, however, that the effect is significantly more pronounced in the model than in the data, suggesting that our model for *S. cerevisiae* transcriptional cell cycle oscillations is missing a mechanism that buffers against the effects of the *∆yox1∆yhp1* double mutation.

While the qualitative agreement obtained in panels A and B suggests that our inferred model indeed captures important aspects of the transcriptional regulation of cell cycle genes, it is clear that it will fail to predict the effect of deletions that unmask post- transcriptional effects. This is illustrated in panel C of the new Figure 6—figure supplement 1 for the case of *rnr1*, a target gene of the transcription factors Swi4 and Mbp1. Both of these DNA-binding proteins are known to act together with the co-regulator Swi6, via the SBF (Swi4-Swi6) and MBF (Mbp1-Swi6) complexes. Our calibration data set demonstrates a significant cell-cycle dependence for the transcription of *swi4*, whereas both *mbp1* and *swi6* show no significant periodic modulation of their transcription rates (see Figure 2—figure supplement 1). Accordingly, our method infers a GRF based almost entirely on regulatory input from Swi4, and predicts that the transcription rate of *rnr1* is essentially constant in a *swi4* mutant strain, as shown in panel C. In contrast, the *swi4* deletion strain of Bean et al.(2005) exhibits an oscillatory expression of *rnr1,* with a delayed peak time and an increased amplitude. As shown by Bean et al.(2005), *rnr1* is in fact redundantly regulated by SBF and MBF, such that only the *swi4-mbp1* double mutant displays an *rnr1* expression that is essentially cell-cycle-independent. The MBF-mediated regulatory input is likely due to the known cyclin-dependent posttranscriptional regulation of the MBF complex (de Bruin et al., 2008).

Taken together, our validation against published data illustrates both the predictive power and the limitations of the GRFs inferred with our method. The quantitative reconstruction of in vivoGRFs is still in its infancy, and we believe our work establishes the first general and systematic GRF inference based on dynamic transcriptome data. In the future, GRF reconstruction could be taken to the next level by combining dynamic transcriptome data with simultaneously measured dynamic information on protein levels and protein localization. Alternatively, or in addition, GRF reconstruction could be further improved by simultaneous inference from DTA data for the wild-type as well as TF knockout strains. Both of these future directions are discussed in the Discussion section.